# Status of the Resistance of *Aphis gossypii* Glover, 1877 (Hemiptera: Aphididae) to Afidopyropen Originating from Microbial Secondary Metabolites in China

**DOI:** 10.3390/toxins14110750

**Published:** 2022-11-01

**Authors:** Ren Li, Shenhang Cheng, Pingzhuo Liang, Zhibin Chen, Yujia Zhang, Pei Liang, Lei Zhang, Xiwu Gao

**Affiliations:** Department of Entomology, China Agricultural University, Beijing 100093, China

**Keywords:** afidopyropen, resistance monitoring, *Aphis gossypii*

## Abstract

The resistance of cotton aphids to various forms of commonly used pesticides has seriously threatened the safety of the cotton production. Afidopyropen is a derivative of microbial metabolites with pyropene insecticide, which has been shown to be effective in the management of *Aphis gossypii*. Several field populations of *Aphis gossypii* were collected from the major cotton-producing regions of China from 2019 to 2021. The resistance of these populations to afidopyropen was estimated using the leaf-dipping method. The LC_50_ values of these field populations ranged from 0.005 to 0.591 mg a.i. L^−1^ in 2019, from 0.174 to 4.963 mg a.i. L^−1^ in 2020 and from 0.517 to 14.16 mg a.i. L^−1^ in 2021. The resistance ratios for all *A. gossypii* populations ranged from 0.03 to 3.97 in 2019, from 1.17 to 33.3 in 2020 and from 3.47 to 95.06 in 2021. The afidopyropen resistance exhibited an increasing trend in the field populations of Cangzhou, Binzhou, Yuncheng, Kuerle, Kuitun, Changji and Shawan from 2019 to 2021. This suggests that the resistance development of the cotton aphid to afidopyropen is inevitable. Therefore, it is necessary to rotate or mix afidopyropen with other insecticides in order to inhibit the development of afidopyropen resistance in field populations.

## 1. Introduction

Afidopyropen is a novel natural product-derived insecticide that acts as a transient receptor for potential vanilloid subtype channel modulators in the chordotonal organs of insects and classified as insecticide Group 9D by the Insecticide Resistance Action Committee (IRAC) [1,2]. Afidopyropen has shown excellent performance against sucking insects, such as Aphididae [3,4,5], Aleyrodidae [6] and Liviidae [7,8]. Afidopyropen disturbs the function of the insect’s chordal organ, causing it to lose its sense of gravity, balance, sound, position and motion. It makes the insects “deaf”, causing a loss in their sense of coordination and direction, and thus, the occurrence of death due to starvation and desiccation [5,9]. Afidopyropen can easily penetrate leaves, and hence presents excellent effectiveness. It also displays little risk to the environment and to humans [10]. Notably, afidopyropen is barely toxic to pollinating insects such as the honeybee *Apis mellifera* Linnaeus, 1758 (Hymenoptera, Apidae), and other beneficial arthropods, such as *Hippodamia convergens* Guérin-Meneville, 1842 (Coleoptera: Coccinellidae), *Harmonia axyridis* (Pallas, 1773) (Coleoptera:Coccinellidae), *Orius insidiosus* (Say, 1832) (Hemiptera: Anthocoridae) and *Aphidoletes aphidimyza* (Rondani, 1847) (Diptera: Cecidomyiidae) [10,11].

The use of afidopyropen has been registered for the control of aphids and whiteflies in a variety of crops in China, including cotton, cucumber, watermelon, wheat, ornamental rose and tomato. Cotton is one of the most important textile crops and plays a key role in the global agricultural economy [12]. However, cotton production is threatened by the cotton aphid *A. gossypii* Glover, 1877 (Hemiptera: Aphididae) throughout their growth cycle. Currently, the resistance of *A. gossypii* to various traditional insecticides (including neonicotinoids, organophosphorus, pyrethroids and carbamates) makes the control of *A. gossypii* more difficult. Some novel insecticides that make it possible to control the damage of *A. gossypii* in the short term have emerged. It is important to accurately understand the development of cotton aphid resistance in each cotton-producing area for the rational application of a novel insecticide.

*A. gossypii* is a cosmopolitan and polyphagous pest that can threaten more than 700 host plants worldwide, including cotton, cucumber, pepper, etc. [13,14,15]. *A. gossypii* can suck plant juice and spread viruses, which seriously restricts crop growth and causes significant economic damage [16,17,18]. The application of chemical insecticides has been the main strategy to control cotton aphids for several decades [19,20,21,22,23,24]. Consequently, the resistance of cotton aphids to many commonly used insecticides cannot be ignored [25,26,27,28]. It is also necessary to find alternatives with novel modes of action that will contribute to delaying the development of cotton aphid resistance.

Consequently, we monitored the afidopyropen resistance in field populations of the cotton aphid collected from the main cotton-producing areas in China from 2019 to 2021. The monitoring data positively contribute to the prediction of the resistance of these field populations to afidopyropen and can optimize the application of this compound, including the application dose, frequency and interval application with other agents with positive biological activity against cotton aphids. These results will be used for the resistance management of cotton aphids.

## 2. Results

### 2.1. Toxicity of Afidopyropen to Aphis gossypii

We pooled bioassay data from all populations and removed the outline to calculate the comprehensive toxicity of afidopyropen each year. The results show that the LC_50_ values were 0.146, 1.057 and 5.030 mg a.i. L^−1^ in 2019, 2020 and 2021, respectively (Table 1). The LC_50_ obtained in 2019 was used as the relative susceptible baseline for cotton aphids to afidopyropen to calculate the resistance ratio in 2020 and 2021, and the pooled data from all cotton aphid populations showed a resistance 7.24 and 34.45 times higher to afidopyropen, respectively (Table 1). The slopes were 0.638, 0.614 and 0.365 in 2019, 2020 and 2021, respectively, showing a downward trend and indicating the susceptible change of the cotton aphid in genetic heterogeneity to afidopyropen.

The LC_50_ values of afidopyropen against *A. gossypii* field populations were 0.009–0.435 mg a.i. L^−1^, 0.174–4.963 mg a.i. L^−1^ and 0.517–14.164 mg a.i. L^−1^ in 2019, 2020 and 2021, respectively (Table 2). In 2019, Cangzhou was the most susceptible population with an LC_50_ value of 0.009 mg a.i. L^−1^ (Table 2), but the LC_50_ value was 2.372 mg a.i. L^−1^ in 2021. In the continuous toxicity testing of afidopyropen to cotton aphids in the same area, the LC_50_ value increased continuously in the Cangzhou, Binzhou, Yuncheng, Kuerle, Yili, Kuitun, Changji and Shawan populations. The average LC_50_ values were 0.149 mg a.i. L^−1^, 1.789 mg a.i. L^−1^ and 3.838 mg a.i. L^−1^ in 2019, 2020 and 2021, respectively (Table 2), and the average LC_50_ values in 2020 and 2021 were significantly higher than those in 2019 (*p* < 0.001) (Figure 1). The increase in LC_50_ values from 2019 to 2021 indicated that the susceptibility of cotton aphid field populations to afidopyropen gradually decreased with the continuous use of afidopyropen.

### 2.2. The Resistance Levels of Aphis gossypii Field Populations to Afidopyropen in 2019–2021

The resistance ratio (RR) of cotton aphid populations to afidopyropen was calculated using the pooled LC_50_ value (0.146 mg a.i. L^−1^) of 2019 as the relative susceptible baseline. The results show that the cotton aphid populations in 2020 developed to be 6.29 to 33.99 times more resistant in 11 of 16 populations, and the 5 remaining populations were susceptible, with an RR only 1.19 to 4.54 times higher (Table 2). In 2021, 16 of the 18 populations evolved to be 5.60 to 97.01 times more resistant. The two other populations in Wusu and Bole became 3.54 and 4.79 times more resistant, respectively, and were susceptible to afidopyropen (Table 2). Kuitun was the most resistant population in 2021 with an RR of 97.01. The RR was continuously upward in Cangzhou, Binzhou, Yuncheng, Kuerle, Kuitun, Changji, Yili and Shawan in our successive resistance monitoring (Table 2). The average RR values were 1.00, 12.25 and 26.29 in 2019, 2020 and 2021, respectively, and were significantly higher in 2020 and 2021 than in 2019 (*p* < 0.001) (Figure 1). According to the concentration–mortality curve, the efficiency of afidopyropen decreased with an increase in cotton aphid field population resistance (Figure 1).

## 3. Discussion

In 2019, afidopyropen was registered in China and has since displayed excellent efficacy in controlling field populations of aphids and whiteflies throughout China [10,29,30]. In the present study, we monitored the resistance evolution of field populations of the cotton aphid, *A. gossypii*, from several major cotton-producing areas across China, and found that the resistance level of *A. gossypii* to afidopyropen increased significantly from 2019 to 2021. The LC_50_ values of these monitored field populations ranged from 0.005 to 0.591 mg a.i. L^−1^ in 2019, from 0.174 to 4.963 mg a.i. L^−1^ in 2020 and from 0.517 to 14.16 mg a.i. L^−1^ in 2021. However, almost all these LC_50_ values were lower than the recommended concentrations of 10–26 mg a.i. L^−1^ of afidopyropen in cotton fields for the control of *A. gossypii*. The LC_50_ values of some populations increased significantly, indicating an increase in their resistance level, but afidopyropen still showed an excellent control effect on *A. gossypii* populations in most cotton areas. Similar results were also found in the control of other insect pests, such as *Aphis glycines* Matsumura (Hemiptera: Aphididae) (0.0013 to 0.40 mg a.i. L^−1^) [3], and whitefly *Bemisia tabaci* Gennadius 1889 (Hemiptera: Aleyrodidae) (1.15 to 4.52 mg a.i. L^−1^) [6].

Following its first registration in Australia, afidopyropen from BASF has been added to the IRAC MoA Classification Scheme as Group 9D, and is currently the only member of this insecticide sub-group [10]. Afidopyropen has a novel mechanism of action that makes it difficult to produce cross-resistance with other registered insecticides. No cross-resistance between afidopyropen and other insecticides was detected. Our previous research showed that these cotton aphid field populations developed 175~56 409 times more resistance to imidacloprid in 2020 [31]. In addition, we found that the resistance of cotton aphids in all field populations to beta-cypermethrin and deltamethrin was more than 10,000 times higher. Afidopyropen is a new biological insecticide derived from natural products that has little negative impact on the ecological environment [2]. Additionally, afidopyropen has low toxicity, which is beneficial to arthropods [10,11]. Notably, we found that afidopyropen showed excellent biological activities against field populations of the cotton aphid. The LC_50_ values of afidopyropen against *A. gossypii* were significantly lower than those reported in previous studies of a potential selective insecticide sulfoxaflor [18,31,32]. These results indicate that afidopyropen can be used as an effective insecticide in controlling *A. gossypii* in China.

To date, several monitored *A. gossypii* field populations have developed resistance to afidopyropen, based on the results of this study. The relative susceptible baseline of *A. gossypii* to afidopyropen was determined based on the susceptibility of field populations in 2019 which can provide a reference standard for the resistance-monitoring programs. A susceptibility baseline of 0.009 mg a.i. L^−1^ of *A. gossypii* to afidopyropen has been reported by Shi et al. [29], but this baseline focused on the most susceptible field population. Based on this baseline, a seemingly higher resistance ratio can be obtained; however, this resistance ratio was irrelevant to the need for field control of the cotton aphid. Therefore, toxicity data of all field populations in 2019 were generated for the dose–effect regression analysis, and the LC_50_ value obtained was used to establish the susceptibility baseline to meet the actual resistance level of most field populations in this study.

Overall, the resistance level of most *A. gossypii* field populations to afidopyropen showed an increasing trend from 2019 to 2021. The resistance ratio for all *A. gossypii* populations ranged from 0.03 to 3.97 in 2019, from 1.17 to 33.3 in 2020 and from 3.47 to 95.06 in 2021. The Kuitun population was the most resistant in 2021 with an RR of 95.06. The resistance levels of the field populations of Cangzhou, Binzhou, Yuncheng, Kuerle, Kuitun, Changji and Shawan increased continuously during continuous resistance monitoring. Therefore, the rotational application of afidopyropen with other insecticides and with different modes of action can be employed to delay the development of cotton aphid resistance in cotton fields [33,34,35]

In this study, the 72 h LC_50_ values were selected as the toxicity endpoint after short-term exposure. Although afidopyropen rapidly inhibits aphid feeding, mortality may be delayed. Therefore, the results from relatively short-duration bioassays may not accurately reflect the actual control effects in the field, especially based on the number of live aphids on plants [9,10]. Solís-Aguilar et al. [8] showed an increased mortality rate of *Diaphorina citri* Kuwayama, 1908 (Hemíptera: Liviidae) nymphs after 7 days. A significant reduction in nymph populations of *B. tabaci* (Gennadius) (Hemiptera: Aleyrodidae) were found after exposure to afidopyropen for three weeks [36]. Consequently, the actual control ability of afidopyropen on the *A. gossypii* field populations will be better than that of the 72 h exposure in this study.

## 4. Conclusions

These results indicate that afidopyropen has excellent potential against resistant populations of the cotton aphid in fields. In addition, the cross-resistance between afidopyropen and other insecticides has not yet been detected. However, we found that several field populations have developed low and moderate resistances to afidopyropen in 2020 and 2021. Therefore, it is essential to continuously monitor the development of the resistance to afidopyropen, reduce the application frequency of afidopyropen and rotate it with the currently available insecticides to avoid cross-resistance.

## 5. Materials and Methods

### 5.1. Aphis gossypii Field Populations

Field populations of *A. gossypii* were collected from the main cotton-producing areas, including Xinjiang, Shandong, Shanxi, Hebei, Hubei and Henan provinces in 2020–2021 from June to September, which was the critical period for cotton growing (cotton boll stage) and when *Aphis gossypii* is most harmful to cotton in China. Xinjiang, as the main cotton-producing area in China, has a higher population density of cotton aphids than other cotton-producing areas in the same period of each year. According to a survey, cotton aphids were present on all cotton plants in Xinjiang at the end of June in 2019, and the number of cotton aphids per 100 cottons was 37,635 [37]. In the middle of July 2020, the number of cotton aphids per 100 cottons reached 94,000 [38]. At present, there is no specific threshold for the control of the cotton aphid. Pesticides are usually applied when there are more than five thousand cotton aphids per 100 cotton plants according to the customs of farm workers.

Pyrethroids have been used to manage cotton aphids in these regions for over 30 years, while neonicotinoids have been used for 20 years. Afidopyrioen has been applied to control cotton aphids since 2019 and generally used no more than five times each year according to BASF’s recommendation. The susceptibility of cotton aphids to afidopyropen was determined in field populations in 2019 [29]. Over 2000 aphids were sampled according to the five-point sampling method on 20–30 cotton plants from main cotton-growing areas (Table 3) and used to establish the population in the laboratory. Cotton aphids were reared on cotton seedlings (*Gossypium hirsutum* L. var. Xinmian No.1). All field populations were reared in insectaria under controlled conditions of 22 ± 1 °C, relative humidity of 60–70% and a photoperiod of 16: 8 h (L: D). Field populations were reared for 3–4 generations in insectaria and used for toxicity tests after the population became stable.

### 5.2. Chemicals

Analytical acetone (>99.5% purity) was obtained from Sino-Pharm Chemical Reagent Co., Ltd. Afidopyropen (97.6%) was obtained from BASF Co., Ltd., (Beijing, China). Triton X-100 was purchased from Sigma-Aldrich Co. (Saint Louis, MO, USA).

### 5.3. Toxicity Bioassays

The toxicity of afidopyropen to field populations was determined by the leaf-dipping method according to a previously described method [39] with slight modifications. The original afidopyropen liquid was obtained by using acetone to configure and be stored in refrigerators at 4 ℃, and the original afidopyropen liquid was used to dilute a series of concentration gradients using 0.05% (*v*/*v*) Triton-X 100 that was configured with distilled water. Fresh cotton leaves without any pesticide exposure were cut into 21 mm diameter leaf discs with a punch, and then these leaf discs were immersed in diluted afidopyropen solutions for 15 s. Leaf discs treated with 0.05% (*v*/*v*) Triton-X 100 water were used as controls. The treated leaf discs were placed indoors to dry, and then the dried leaf discs were placed in 12-well cell plates which contained 2.5 mL of 1.85% (*w*/*v*) agar. Healthy apterous adult aphids were gently transferred from cotton seedlings into 12-well cell plates using a soft small brush and then the plates were sealed with Chinese art paper to prevent aphids from escaping, with 3 replicates per concentration and 25–30 aphids in each well. The 12-well cell plates were placed under the same condition as the aphids’ rearing. The number of live and dead aphids was recorded after 72 h. Aphids that were unable to move under the touch of a small soft brush were considered dead.

### 5.4. Statical Analysis

The values of LC_50_ and LC_90_, the 95% confidence limits (CLs), *χ*^2^ and degree of freedom (*df*) and the slope of the regression curve were calculated by Polo plus 2.0 software (www.LeOra-Software.com, accessed on 20 September 2022). Combining all the populations’ bioassay data and removing the outline calculated the comprehensive LC_50_ and LC_90_, the 95% confidence limits (CLs), *χ*^2^ and degree of freedom and the slope of the regression curve at each year using Polo plus 2.0 software. The comprehensive LC_50_ in 2019, by using the published data [29], was rebuilt and used as a relative susceptible baseline of cotton aphid to afidopyropen in China to calculate the resistance ratio (RR), wherein 0 < RR ≤ 5 was considered as susceptible, 5 < RR ≤ 10 as a low resistance level, 10 < RR ≤ 100 as a moderate resistance level and RR > 100 as high resistance level. The comparison of LC_50_ and resistance ratio (RR) each year was completed by the one-way ANOVA analysis using GraphPad Prism 7.0 (GraphPad Software Inc., San Diego, CA, USA). The geographical map was drawn using ArcMap GIS10.2 software (Environment System Research Institute, Redlands, CA, USA). The concentration–mortality curve was finished by R 4.0.5 (https://www.r-project.org/, accessed on 31 March 2021).

## Figures and Tables

**Figure 1 toxins-14-00750-f001:**
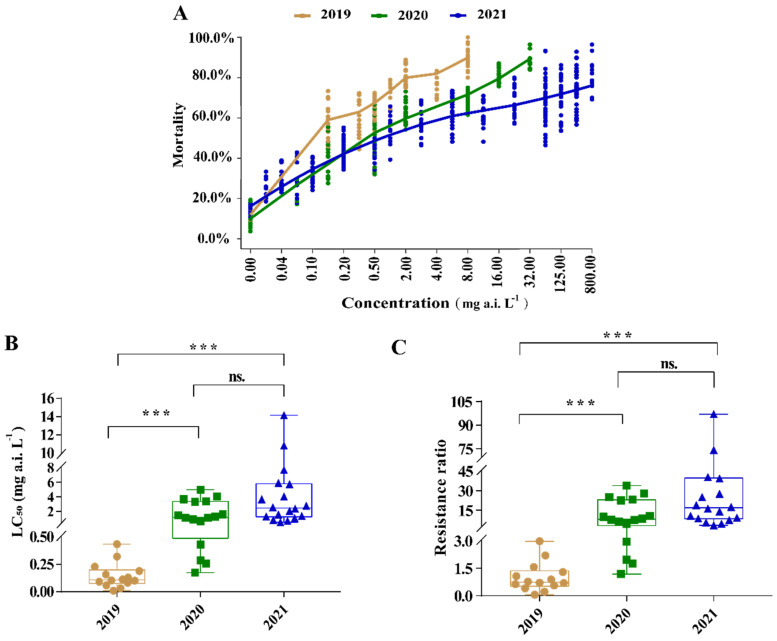
The comparison of the afidopyropen toxicity to *Aphis. gossypii* from 2019 to 2021. (**A**): The concentration–mortality curve was completed by combining all mortality rates with each concentration in each year from 2019 to 2021. (**B**): The comparison of the pool value of LC_50_ at each year, “***” and “ns.” indicate an extremely different significance and no significance, respectively. (**C**): The comparison of the pool value of resistance ratio at each year, “***” and “ns.” indicate extremely different significance and no significance, respectively.

**Table 1 toxins-14-00750-t001:** The comprehensive toxicity of afidopyropen to *Aphis gossypii* populations in each year of 2019–2021.

Year	Slope ± SE ^a^	LC_50_ (95% CL) ^b^ mg a.i. L^−1^	LC_90_ (95% CL) ^b^ mg a.i. L^−1^	RR ^c^	*χ*^2^ (*df*) ^d^	*p* Value
2019	0.638 ± 0.042	0.146 (0.103~0.195)	15.21 (10.68~24.49)	1.00	102.24 (122)	0.90
2020	0.614 ± 0.033	1.057 (0.829~1.317)	128.9 (85.4~212.0)	7.24	108.53 (117)	0.70
2021	0.365 ± 0.014	5.030 (3.835~6.568)	16,420 (9102~32,254)	34.45	206.93 (294)	1.00

^a^ SE: Standard error; ^b^ 95% CL: 95% Confidence limit; ^c^ RR: Resistance ratio = LC_50_ of comprehensive population/LC_50_ of 2019; ^d^ *χ*^2^ (*df*): Chi-squared (*χ*^2^), and degrees of freedom (*df*).

**Table 2 toxins-14-00750-t002:** The toxicity of afidopyropen to *Aphis gossypii* field populations in 2019–2021.

Year	Province	City	*n* ^a^	Slope ± SE ^b^	LC_50_ (95% CL) ^c^ mg a.i. L^−1^	RR ^d^	*χ*^2^ (*df*) ^e^	*p*
2019 [29]	Hebei	Cangzhou	414	0.95 ± 0.2	0.009 (0.000~0.024)	0.06	19.11 (11)	0.06
	Hebei	Hengshui	579	0.52 ± 0.15	0.229 (0.002~0.787)	1.57	17.18 (15)	0.31
	Shandong	Binzhou	444	0.62 ± 0.11	0.101 (0.022~0.243)	0.69	11.61 (13)	0.56
	Shandong	Dongying	558	0.50 ± 0.12	0.321 (0.073~1.323)	2.20	22.92 (16)	0.12
	Shandong	Dezhou	512	0.63 ± 0.15	0.156 (0.032~0.323)	1.07	11.28 (14)	0.66
	Shandong	Heze	447	1.61 ± 0.28	0.112 (0.065~0.175)	0.77	12.56 (11)	0.32
	Shanxi	Yuncheng	503	0.57 ± 0.12	0.435 (0.211~1.072)	2.98	13.89 (16)	0.61
	Xinjiang	Akesu	468	0.44 ± 0.16	0.058 (0.000~0.266)	0.40	12.77 (16)	0.69
	Xinjiang	Bole	521	0.81 ± 0.14	0.189 (0.078~0.325)	1.29	15.13 (16)	0.52
	Xinjiang	Kuerle	582	0.78 ± 0.14	0.092 (0.012~0.221)	0.63	22.06 (16)	0.14
	Xinjiang	Nongwushi	423	0.70 ± 0.16	0.082 (0.003~0.236)	0.56	15.69 (12)	0.21
	Xinjiang	Shihezi	486	1.02 ± 0.18	0.130 (0.041~0.243)	0.89	9.23 (14)	0.82
	Xinjiang	Tulufan	402	0.44 ± 0.1	0.104 (0.022~0.242)	0.71	3.34 (11)	0.99
	Xinjiang	Wusu	616	0.84 ± 0.19	0.032 (0.001~0.107)	0.22	15.36 (15)	0.43
2020	Hebei	Hengshui	481	0.77 ± 0.12	1.448 (0.677~2.529)	9.92	5.35 (11)	0.91
	Shandong	Binzhou	568	0.56 ± 0.09	0.663 (0.256~1.267)	4.54	5.46 (11)	0.91
	Shandong	Dongying	428	0.44 ± 0.1	0.287 (0.025~0.959)	1.97	5.27 (11)	0.92
	Shandong	Xiajin	455	0.42 ± 0.1	0.431 (0.036~1.612)	2.95	3.16 (12)	0.99
	Shanxi	Yuncheng	483	0.92 ± 0.13	1.119 (0.295~2.248)	7.66	20.78 (12)	0.05
	Xinjiang	Alaer	615	0.62 ± 0.09	1.128 (0.567~1.974)	7.73	10.41 (13)	0.66
	Xinjiang	Bole	427	0.97 ± 0.19	4.963 (2.765~8.621)	33.99	8.98 (10)	0.53
	Xinjiang	Changji	460	0.56 ± 0.11	0.174 (0.016~0.522)	1.19	11.32 (11)	0.42
	Xinjiang	Kuerle	383	1.03 ± 0.15	1.592 (0.752~2.754)	10.90	10.79 (11)	0.46
	Xinjiang	Kashi	459	1.08 ± 0.14	3.658 (2.463~5.323)	25.05	10.18 (12)	0.60
	Xinjiang	Kuitun	517	0.66 ± 0.09	1.250 (0.709~2.020)	8.56	5.57 (11)	0.90
	Xinjiang	Shihezi	449	1.23 ± 0.16	3.303 (2.184~4.937)	22.62	9.42 (11)	0.58
	Xinjiang	Shawan	543	0.84 ± 0.09	0.919 (0.579~1.386)	6.29	5.57 (13)	0.96
	Xinjiang	Tulufan	462	0.69 ± 0.12	3.380 (1.721~5.508)	23.15	5.39 (11)	0.91
	Xinjiang	Wusu	459	1.05 ± 0.13	4.045 (2.463~5.323)	27.71	10.18 (12)	0.60
	Xinjiang	Yili	499	0.78 ± 0.09	0.256 (0.131~0.419)	1.75	2.97 (11)	0.99
2021	Hebei	Cangzhou	585	0.50 ± 0.06	2.372 (0.796~6.396)	16.25	16.97 (13)	0.20
	Hebei	Hengshui	548	0.41 ± 0.06	0.817 (0.232~2.303)	5.60	16.51 (14)	0.28
	Henan	Nanyang	582	0.35 ± 0.08	5.740 (1.408~24.784)	39.32	13.16 (14)	0.51
	Shandong	Binzhou	609	0.55 ± 0.09	1.562 (0.618~3.379)	10.70	3.9 (12)	0.99
	Shanxi	Yuncheng	616	0.41 ± 0.07	10.821 (3.640~31.297)	74.12	7.34 (15)	0.95
	Xinjiang	Alaer	615	0.61 ± 0.07	2.552 (1.241~4.727)	17.48	10.37 (13)	0.66
	Xinjiang	Bole	723	0.42 ± 0.06	0.700 (0.234~1.681)	4.79	9.21 (14)	0.82
	Xinjiang	Changji	650	0.66 ± 0.1	5.898 (2.344~15.991)	40.40	13.53 (12)	0.33
	Xinjiang	Kuche	583	0.33 ± 0.07	1.363 (0.144~4.926)	9.34	12.28 (12)	0.42
	Xinjiang	Kuerle	610	0.53 ± 0.07	7.688 (3.286~16.608)	52.66	12.25 (14)	0.59
	Xinjiang	Kuitun	652	0.79 ± 0.12	14.164 (7.057~27.439)	97.01	11.35 (14)	0.66
	Xinjiang	Shihezi	626	0.28 ± 0.07	2.76 (0.287~13.613)	18.90	4.18 (11)	0.96
	Xinjiang	Shawan	590	0.5 ± 0.08	1.285 (0.196~4.279)	8.80	16.39 (11)	0.13
	Xinjiang	Shaya	576	0.4 ± 0.07	4.044 (0.502~17.058)	27.70	15.68 (13)	0.27
	Xinjiang	Tulufan	582	0.61 ± 0.09	1.095 (0.359~2.705)	7.50	14.53 (12)	0.27
	Xinjiang	Tumushuke	605	0.42 ± 0.07	3.657 (0.894~10.993)	25.05	5.89 (11)	0.88
	Xinjiang	Wusu	585	0.42 ± 0.06	0.517 (0.130~1.496)	3.54	8.84 (13)	0.78
	Xinjiang	Yili	592	0.52 ± 0.07	2.048 (1.579~5.686)	14.03	18.53 (14)	0.18

^a^ n: Numbers of use toxicity testing; ^b^ SE: Standard error; ^c^ 95% CL: 95% Confidence limit; ^d^ RR: Resistance ratio = LC_50_ of field population/0.146 mg a.i. mL^−1^; ^e^ *χ*^2^ (*df*): Chi-squared (*χ*^2^), and degrees of freedom (*df*).

**Table 3 toxins-14-00750-t003:** The location of *Aphis gossypii* collecting sites from 2019–2021.

Year	Location (City, Province)	Longitude and Latitude	Hosts
2020	Aaler, Xinjiang	81.28° E, 40.55° N	Cotton
	Bole, Xinjiang	82.05° E, 44.85° N	Cotton
	Binzhou, Shandong	118.02° E, 37.43° N	Cotton
	Changji, Xinjiang	87.31° E, 44.01° N	Cotton
	Dongying, Shandong	118.58° E, 37.45° N	Cotton
	Hengshui, Hebei	115.58° E, 37.55° N	Cotton
	Kuerle, Xinjiang	86.17° E, 41.73° N	Cotton
	Kashi, Xinjiang	75.99° E, 39.47° N	Cotton
	Kuitun, Xinjiang	84.9° E, 44.43° N	Cotton
	Shihezi, Xinjiang	86.08° E, 44.31° N	Cotton
	Shawan, Xinjiang	85.62° E, 44.33° N	Cotton
	Tulufan, Xinjiang	89.19° E, 42.94° N	Cotton
	Wusu, Xinjiang	84.68° E, 44.44° N	Cotton
	Xiajin, Shandong	116.00° E, 36.95° N	Cotton
	Yuncheng, Shanxi	111.00° E, 35.02° N	Cotton
	Yili, Xinjiang	81.32° E, 43.92° N	Cotton
2021	Alaer, Xinjiang	81.24° E, 40.56° N	Cotton
	Bole, Xinjiang	82.05° E, 44.85° N	Cotton
	Binzhou, Shandong	118.02° E, 37.43° N	Cotton
	Changji, Xinjiang	87.31° E, 44.01° N	Cotton
	Cangzhou, Hebei	116.87° E, 38.31° N	Cotton
	Hengshui, Hebei	115.58° E, 37.55° N	Cotton
	Kuche, Xinjiang	83.05° E, 42.08° N	Cotton
	Kuerle, Xinjiang	86.39° E, 40.59° N	Cotton
	Kuitun, Xinjiang	84.90° E, 44.43° N	Cotton
	Nanyang, Henan	112.54° E, 33.00° N	Cotton
	Shihezi, Xinjiang	86.08° E, 44.31° N	Cotton
	Shawan, Xinjiang	85.62° E, 44.33° N	Cotton
	Shaya, Xinjiang	82.92° E, 41.25° N	Cotton
	Tulufan, Xinjiang	89.19° E, 42.94° N	Cotton
	Tumushuke, Xinjiang	79.21° E, 40.00° N	Cotton
	Wusu, Xinjiang	84.68° E, 44.44° N	Cotton
	Yuncheng, Shanxi	111.00° E, 35.02° N	Cotton
	Yili, Xinjiang	81.32° E, 43.92° N	Cotton

## Data Availability

The data presented in this study are available in this article.

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
