# Peer review of "Status of the Resistance of Aphis gossypii Glover, 1877 (Hemiptera: Aphididae) to Afidopyropen Originating from Microbial Secondary Metabolites in China"

_toxins, 2022, doi:10.3390/toxins14110750_

Round 1

Reviewer 1 Report

The manuscript presents important information that deserves to be published in Toxins. However, before acceptance, I suggest the modifications below. Note: all citations and references must be rearranged in ascending numerical order (0, 1, 2, 3...) (as the material and methods are at the end, the references cited in the methodology must be with the last numbers).

Line 1

Change “Aphis gossypii Glover” for “Aphis gossypii Glover, 1877 (Hemiptera: Aphididae)”

Line 7 and 8

Change “Aphis gossypii Glover” for “Aphis gossypii

Change “Aphis gossypii” for “Aphis gossypii

Lines 30-33

Change "Apis mellifera (Hymenoptera: Apidae)" for "Apis mellifera Linnaeus, 1758 (Hymenoptera, Apidae)"

Change “Hippodamia convergens (Coleoptera: Coccinellidae),” for “Hippodamia convergens Guérin-Meneville, 1842 (Coleoptera: Coccinellidae),”

Change “Harmonia axyridis (Coleoptera: Coccinellidae)” for “Harmonia axyridis (Pallas, 1773) (Coleoptera:Coccinellidae)”

Change “Orius insidiosus (Hemiptera: Anthocoridae)” for “Orius insidiosus (Say, 1832) (Hemiptera: Anthocoridae)”

Change “Aphidoletes aphidimyza (Diptera: Cecidomyiidae)” for “Aphidoletes aphidimyza (Rondani, 1847) (Diptera: Cecidomyiidae)”

Line 38

Change ““A. gossypii” for “A. gossypii Glover, 1877 (Hemiptera: Aphididae)”

Line 44

Change “A. gossypii Glover (Hemiptera: Aphididae)” for “A. gossypii

Line 46

Change “A. gossypii” for “Aphis gossypii

Line 64, 68, 71

Tables must be numbered in ascending order

Change “(Table 2)” for “(Table 1)”

Line 79, 80, 84, 94, 95, 97, 102

Change “(Table 3)” for “(Table 2)”

Line 128

Change “Bemisia tabaci” for “Bemisia tabaci Gennadius 1889 (Hemiptera: Aleyrodidae)”

Line 147

Change “Shi et al. (2022) [29]” for “Shi et al. [29]”

Line 165

Change “Solís-Aguilar et al. (2015)” for “Solís-Aguilar et al. [8]”

Line 166

Change “Diaphorina citri (Hemíptera: Liviidae) “ for “Diaphorina citri Kuwayama, 1908 (Hemíptera: Liviidae)”

Line 179

5. Materials and methods - references should be cited in ascending numerical order

Line 189, 194

Change “(Table 1)” for “(Table 3)”

Reviewer 2 Report

The manuscript “Status of the resistance of Aphis gossypii Glover to afidopyropen originating from microbial secondary metabolites in China” is written well, results are well illustrated, the discussion is nice to read. But M&M written not clear. There are few comments to the authors. 

1. Introduction. Please, give explanation about origin of novel natural product-derived insecticide Afidopyropen (Group 9D). What does it mean Group 9D, is it world classification? From what organism it was extracted? What the nature of the insecticide – protein, lipid or what?

Also, please add information about regalements of application, how many times in the season Afidopyropen could be used, and how long period between treatments.

2. M&M. 5.1. line 187. “Over 2000 aphids were sampled at each location”. It is not clear for the readers, please, clarify one field population – it is a population from one province or from one City? Each location - it is a population from one province or from one City?

3. M&M. Table 3. Please explain to the readers, why did authors take insects from different number of cities from one province in different year? For example, Hebei – 2 cities in 2019; 1 city in 2020, 3 cities in 2021.

4. M&M. Did authors know the population density at each field at each year? Was it harmfulness threshold on each field? It is critical because susceptibility to the insecticides will be different in the population peak and outgrowth.

5. M&M. Did authors know how many times Afidopyropen were applied in the field before the insects were collected? Authors pooled bioassay data, and the differences in the background of the populations may affect the results in this manuscript.

6. Discussion. Line 133-134. “No cross- 133 resistance between afidopyropen and other insecticides was detected.” Is it the literature data or authors experiments? If literature data, please add the reference. 

Round 2

Reviewer 2 Report

Thank you for the responses.

4. M&M. Did authors know the population density at each field at each year? Was it harmfulness threshold on each field? It is critical because susceptibility to the insecticides will be different in the population peak and outgrowth

Response:
Thanks for your comments. It is the critical period for cotton growing (cotton boll stage) and is most harmful of Aphis Gossypii to cotton from June to September in China. Xinjiang, as the main cotton producing area in China, has a higher population density of cotton aphids than other cotton producing areas in the same period of each year. According to survey, cotton aphids were present on all cotton plants in Xinjiang at the end of June in 2019, and the number of aphids per 100 cottons was 37 635 [1]. In the middle of July 2020, the number of cotton aphids per 100 cottons reached 94 000 [2]. At present, there is no specific threshold for the control of the cotton aphid. Pesticides are usually applied when there are more than five thousand cotton aphids per 100 cotton plants according to the custom of farm workers.

Thank you for the response and please, add this information to the MS.

5. M&M. Did authors know how many times Afidopyropen were applied in the field before the insects were collected? Authors pooled bioassay data, and the differences in the background of the populations may affect the results in this manuscript.

Response: Thanks for your comments. Afidopyrioen are generally used no more than five times each year according to BASF's recommendation.

The cotton aphid field populations were usually collected after the once application of afidopyropen in 2020 and 2021. – please, add this information to the MS, readers need it.
